# Non-Canonical Functions of Myeloperoxidase in Immune Regulation, Tissue Inflammation and Cancer

**DOI:** 10.3390/ijms232012250

**Published:** 2022-10-14

**Authors:** Joey S. Lockhart, Ronen Sumagin

**Affiliations:** Department of Pathology, Feinberg School of Medicine, Northwestern University, 300 East Superior St., Chicago, IL 60611, USA

**Keywords:** neutrophil, trafficking, inflammation, cancer, endothelial cell, therapeutic, reactive oxygen species, cell activation, tissue injury

## Abstract

Myeloperoxidase (MPO) is one of the most abundantly expressed proteins in neutrophils. It serves as a critical component of the antimicrobial defense system, facilitating microbial killing via generation of reactive oxygen species (ROS). Interestingly, emerging evidence indicates that in addition to the well-recognized canonical antimicrobial function of MPO, it can directly or indirectly impact immune cells and tissue responses in homeostatic and disease states. Here, we highlight the emerging non-canonical functions of MPO, including its impact on neutrophil longevity, activation and trafficking in inflammation, its interactions with other immune cells, and how these interactions shape disease outcomes. We further discuss MPO interactions with barrier forming endothelial and epithelial cells, specialized cells of the central nervous system (CNS) and its involvement in cancer progression. Such diverse function and the MPO association with numerous inflammatory disorders make it an attractive target for therapies aimed at resolving inflammation and limiting inflammation-associated tissue damage. However, while considering MPO inhibition as a potential therapy, one must account for the diverse impact of MPO activity on various cellular compartments both in health and disease.

## 1. Introduction

Myeloperoxidase (MPO) is a heme peroxidase enzyme predominantly expressed by polymorphonuclear neutrophils and to a much lesser degree monocytes and macrophage subsets. MPO is a major component of the anti-microbial defense system, facilitating the killing of ingested pathogens within the neutrophil phagosome [1]. MPO constitutes up to 5% of the dry cellular mass in activated neutrophils [2,3], consistent with its critical role as a neutrophil effector of host defense. This high molecular mass (~150 kDa) protein is produced in a sequential process and stored within the primary azurophilic granules of neutrophils until activation [4]. Upon neutrophil stimulation (for example, in response to phagocytosis of pathogenic bacteria), the majority of MPO is sequestered to the phagosome, however some of it may escape into the local inflamed environment, impacting the function of neutrophils and neighboring cells [5,6,7,8].

Given its inherent cytotoxicity, MPO escape into the surrounding tissue can lead to substantial cellular damage and exacerbate inflammation. There are several reports that illustrate the ability of MPO to cause extensive tissue damage after it is released from activated neutrophils (recently reviewed by [9,10]). The generation of MPO is tightly linked to neutrophil maturation in the bone marrow. As promyelocytes transition to become mature neutrophils, many proteolytic enzymes, including MPO are segregated in a tightly regulated sequential process into distinct granular compartments, to be released upon specific cues [11]. MPO is strictly localized to the azurophilic or primary granules, which are formed first in developing neutrophils at the promyelocyte stage [12]. In addition to MPO, azurophilic granules contain several other proteolytic enzymes, such as lysozyme, cathepsins, defensins and neutrophil elastase (NE), all of which play important roles in neutrophil effector function [13].

In maturing neutrophils, MPO synthesis begins with the generation of an inactive precursor known as apoproMPO [14]. ApoproMPO interacts with several chaperone molecules in the ER that allow for the addition of an iron protoporphyrin IX (heme) to generate proMPO [15]. Further processing produces the enzymatically active MPO molecule consisting of a homodimer with monomers composed of heavy and light chains, and with each heavy chain covalently linked to a heme group [14,16].

Upon neutrophil activation, an NADPH oxidase (NOX) complex, specifically NOX2 associates with the membrane of the phagosome to catalyze the transfer of electrons from cytoplasmic NADPH to molecular oxygen, giving rise to superoxide radical anions [17]. To limit extensive damage within the cell, superoxide is rapidly converted to hydrogen peroxide and oxygen, either spontaneously or via the action of superoxide dismutase [18]. MPO interacts with hydrogen peroxide to generate some of the strongest antimicrobial compounds, such as hypochlorous acid (HOCl), reactive halide species HOBr and HOSCN, and singlet oxygen [1,19]. Reactive oxidants generated by MPO within the phagosome exert their antimicrobial effects to provide host immunity [20,21,22,23].

Given the important function of MPO in host defense, its mode of action and the anti-bacterial activity has been extensively studied. Interestingly, in additional to its canonical function within the phagosome, MPO has been shown to directly bind select pathogens, and this has been proposed as a mechanism to locally target the activity of reactive oxidants to limit collateral tissue damage [24]. This was demonstrated to be due to the rapid consumption of H_2_O_2_ via the catalytic activity of MPO and others provide evidence for a similar protective effect of MPO on H_2_O_2_-induced cytotoxicity in murine macrophages [24,25]. Conversely, MPO can be self-degraded through the action of its catalytic products to provide a source of free iron, which can cause tissue damage via the formation of additional ROS and increase the risk of proliferation of pathogenic microbes [26]. Free iron can generate significant levels of toxic reactive oxidants through interactions with hydrogen peroxide, a process commonly referred to as the Fenton reaction. The formation of chloramines due to HOCl and other MPO oxidative products significantly impacts protein activity and structure (recently reviewed by [27]). Since proteins are a major target for HOCl in biological systems, 3-chlorotyrosine is often used as an effective marker of MPO activity [27,28,29,30,31]. MPO products have also been shown to impact the activity of other proteins. For example, HOCl can interact with zinc-histidine/cysteine clusters, impacting protein structure and disrupting enzymatic activity of yeast alcohol dehydrogenase [32].

Neutrophil extracellular trap (NET) formation is perhaps another intriguing non-canonical function of MPO [33]. NETs have been described as a mechanism of trapping and killing of invading pathogens, as well as promoting recruitment of other immune cells to combat microorganisms. NET formation follows nuclear DNA decondensation and requires the presence of MPO, neutrophil elastase (NE), and peptidylarginine deiminase type IV (PAD-4) [34,35]. How MPO activity mediates NETs release is not completely understood, but its presence on NETs themselves likely contributes to local ROS generation and killing of trapped pathogens [36]. Interestingly, recent work demonstrates the direct inhibition of NET-associated MPO by muscle-derived mesenchymal stem cells (MSCs) [37]. MSCs are known to elicit significant immunomodulatory effects, including the ability to limit excessive inflammation (reviewed by [38,39,40]). Franck and colleagues demonstrated that MSCs inhibit NET-MPO even after the release of NETs from activated neutrophils, indicating that MSCs may have therapeutic potential to treat autoimmune diseases or inflammatory conditions that are driven by the catalytic activity of MPO [37,41].

MPO is a well-established neutrophil marker and consistent with many pathological roles of neutrophils in inflammatory diseases and with the fact that although well-contained, it can leak into surrounding tissue, elevated MPO serum levels are found in several inflammatory conditions [42,43,44,45]. Despite frequent detection of partial or complete MPO deficiency in the population (~1:4000), in many cases patients experience no complications and can clear most bacterial, fungal, or viral infections [20,46]. This suggests redundancy (as one would expect for such critical function) or an induction of compensatory antimicrobial mechanisms in mammalian hosts. Clinical data also suggest that in some cases MPO-deficient individuals take longer to clear certain pathogens such as *S. aureus* and often experience recurring fungal infections with organisms such as *C. albicans* [47,48]. The antimicrobial activity of MPO is therefore an important aspect of healthy immune function, but evidence emerges for additional, regulatory roles for MPO.

Neutrophil hematopoiesis requires a substantial energy investiture, as ~2 × 10^11^ neutrophils are produced by the bone marrow every day. This roughly amounts 60% of the total energy devoted to hematopoiesis [11]. Synthesis of MPO protein, being ~5% of the dry mass of neutrophils, thus similarly represents a significant metabolic investment [3]. This highlights the likelihood of important regulatory functions of MPO, both canonical and non-canonical in mammalian hosts. In the following sections we will discuss the emerging non-canonical functions of MPO, including regulating neutrophil trafficking, signaling, interactions with other immune cells and its role in inflammatory diseases and cancer.

## 2. MPO and Inflammatory Disorders

As we have discussed above, neutrophils are a major source of MPO, and although most of it is contained within neutrophil granules or the lysosome, MPO release and accumulation in tissue, for example, endothelial cells has been observed [49,50,51]. Several mechanisms aiding in MPO dissemination to various tissue compartments have been described. For example, MPO released by circulating neutrophils can reach distal organs via binding to erythrocytes [52,53] or by being transported by plasma proteins such as albumin and various lipoproteins [49,54]. The interplay between MPO and proteins are likely mediated via electrostatic interactions due to the polycationic nature of the molecule. Ceruloplasmin can bind MPO and inhibit its catalytic activity, potentially contributing to MPO transport in the circulation [55]. MPO can also be shuttled by extracellular vesicles (EVs) or microparticles (MPs) to mediate potent effects on surrounding cells [5,56]. MPO localization to EVs is intriguing, especially given that fact that it requires a specific stimulus leading to mobilization and release of primary granules. These interactions allowing for the dissemination of MPO are thus likely potentiating the pathologic impact of MPO in inflammation.

MPO is a common inflammatory marker, and elevated MPO levels are observed in many pathological conditions, including inflammatory bowel disease (IBD) [45,57,58,59], cardiovascular disease [60,61,62,63], obesity [43,64,65], liver disease [66,67], arthritis [68], multiple sclerosis [69,70], Alzheimer’s disease [71], stroke [72] and cancer [73,74] (Table 1). MPO levels are elevated in IBD, and MPO inhibition in animal models of colitis is protective [58,75,76]. Similarly, MPO inhibition improves colonic wound healing, indicating an important detrimental role of MPO in colon inflammation [5]. In cardiovascular disease, MPO oxidizes low density lipoproteins, promoting the formation of atherosclerotic lesions and potentially contributing to plaque rupture [77,78]. MPO inhibition improves plaque stability in models of atherosclerosis and has been suggested as a target to prevent plaque rupture [79]. In the lungs, MPO is used as a marker for systemic inflammation in smokers, and elevated MPO levels are observed in patients with chronic obstructive pulmonary disease [80,81]. MPO inhibition ameliorates oxidative stress and decreases morbidity in a murine model of cystic fibrosis-related infection [82]. These few examples help illuminate the broad-spectrum of MPO activity in heath and disease. MPO also plays important roles in neuroinflammation and cancer, which we will elaborate upon in the following sections.

Interestingly in the context of various inflammatory diseases, MPO function is mainly considered as detrimental since the relative abundance of MPO positively correlates with disease severity [83]. This is consistent with MPO being a neutrophil marker and the evidence of neutrophil pathologic functions in disease. However, some reports implicate MPO activity as protective in specific disease settings [24,84,85,86]. Thus, while MPO plays a critical role in host defense via its antimicrobial activity, its emerging non-canonical roles both ROS-generation dependent and independent may play important roles in disease pathogenesis and progression. For example, MPO is protective in murine models of LPS-induced inflammation [87]. Mortality, hypothermia and proinflammatory cytokine release after LPS administration are increased in MPO KO mice compared with WTs, suggesting that MPO is beneficial in cases of sepsis [87]. Additionally, MPO may protect surrounding cells from excessive damage by effectively targeting reactive oxidant release specifically to the area of colonizing intracellular pathogens such as *Salmonella* [24]. Authors of this study demonstrated that MPO KO mice released high amounts of H_2_O_2_, resulting in significant tissue damage that is perpetuated by other immune effectors such as macrophages [24]. On the other hand, inhibition of MPO with a pharmacological agent (AZM198) attenuated degranulation, inflammation, and neutrophil-mediated endothelial damage in patients with antineutrophil cytoplasmic antibody (ANCA)-associated vasculitis [88]. These contrasting studies from different inflammatory disorders highlight the fact that MPO can be beneficial or detrimental in the development and progression of inflammation. The various impacts and the non-canonical functions associated with the MPO activity in inflammatory disorders are summarized in Table 1.

**Table 1 ijms-23-12250-t001:** Pathological roles of MPO in tissue inflammation and disease.

Disease	MPO Activity	References
Cardiovascular Disease	Elevated MPO in atherosclerosis associated with disease severity MPO oxidizes low density lipoproteins, promoting the formation of atherosclerotic lesions MPO binds the ECM of smooth muscle cells and alters ECM composition MPO alters the rigidity of platelets, promoting aggregation and vascular inflammation	[60,77,89,90,91,92]
Neurodegenerative Disorders	MPO modulates neuronal, microglial cells, and astrocyte activity MPO-derived ROS cause vascular inflammation in the brain MPO accumulation is observed Alzheimer’s disease and contributes to severity of symptoms (demonstrated via improved health in KO mice) MPO levels positively correlate with MS severity	[69,70,71,93,94,95]
Cancer	MPO induces DNA damage and promotes mutational burden MPO enhances breast cancer progression in murine models MPO converts procarcinogens into carcinogens	[96,97,98]
Inflammatory Bowel Disease	Elevated MPO levels associated with increased morbidity MPO levels are higher in the stool of patients with active IBD MPO impairs resolution of mucosal injury via suppression of IEC migration and proliferation MPO inhibition limits severity of colitis MPO released on NETs exacerbates colitis, promotes tissue damage, and is positively correlated with severity of Crohn’s disease	[42,57,75,76,99]
Obesity	Obese individuals have significantly increased MPO in serum MPO participates in the regulation of obesity, however the specific function remains to be determined MPO inhibition attenuates liver damage associated with obesity	[65,100,101]
Pulmonary Inflammation	MPO degrades the glycocalyx, which is important in neutrophil adhesion and lung injury MPO deficiency promotes or suppresses inflammation in the lung in different models of pulmonary infection MPO inhibition shows limited efficacy as a potential therapeutic for tuberculosis	[82,102,103,104]
Arthritis	Experimentally induced arthritis is attenuated in MPO deficient individuals Elevated MPO levels promote oxidative stress in patients with arthritis	[68,105,106]
Ischemic Stroke	MPO levels positively correlate with the risk for recurrent acute stroke MPO inhibition is protective against inflammation-associated cell death following stroke	[107,108]

## 3. Exogenous MPO Regulates Neutrophil Activation and Trafficking

Critical effector functions of neutrophils take place following activation and crossing of the vascular barrier as they navigate their way to sites of inflammation in the tissue. Regulatory mechanisms of neutrophil activation, including their interactions with barrier forming endothelial and epithelial cells have been extensively studied both in the bloodstream and tissues [109,110,111,112]. However, new molecular mechanisms for these vital processes are still emerging. This includes intriguing evidence of the roles of exogenous MPO in these processes.

Exogenous MPO has been suggested to mimic the action of proinflammatory cytokines by promoting activation of NF-κβ and MAPK signaling, leading to mobilization of intracellular calcium stores and neutrophil degranulation [113,114,115]. MPO can directly bind CD11b on human neutrophils, leading to nuclear translocation of NF-κβ, increased surface expression of CD11b and the release of chemokines and cytokines that promote inflammation [113]. In contrast, neutrophils from MPO KO mice showed an increased ability to engulf zymosan particles in vitro [116]. The authors found that zymosan-stimulated neutrophils from MPO KO animals had elevated surface expression of CD11b compared with WT animals, and that this increased expression was due to enhanced FAK and ERK pathway activity in MPO-deficient animals, leading to increased phagocytic ability [116]. Additional studies on zymosan-stimulated neutrophils demonstrate that MPO-deficiency promotes production of proinflammatory cytokines such as MIP-2, MIP-1α, MIP-1β, IL-1α, IL-1β and TNF-α in vitro and in vivo [85,117]. Thus, it appears that under certain conditions MPO may serve suppressive roles in neutrophil activation. Conversely, others demonstrate that MPO-deficiency attenuates LPS-induced acute lung inflammation and negatively regulates proinflammatory cytokine production [118]. This contrasting evidence suggests a functional dualism for MPO in both suppressing or activating neutrophil effector functions and that this likely depends on the experimental model or timing. Therefore, further research is warranted to uncover the specific role of exogenous MPO in neutrophil activation.

Interestingly, similar controversial evidence suggests that MPO can either delay neutrophil apoptosis (independently of its catalytic activity) [119] or promote apoptosis by enhancing surface phosphatidylserine (PS) expression [120]. In apoptotic neutrophils, MPO has been shown to co-localize with PS at the cell surface [121], suggesting a potential contribution to programmed cell death, but the role of MPO in the regulation of apoptosis remains undefined. Furthermore, MPO release correlates with the number of apoptotic cells in human patients undergoing dialysis, providing additional evidence that MPO is an important regulator of apoptosis [122]. The association of MPO with PS at the cell surface may also have implications for macrophage recruitment and subsequent clearance of apoptotic cells, therefore additional studies that investigate neutrophil-MPO-macrophage interactions would be an interesting avenue to explore.

After sensing inflammatory cues, neutrophils activate their migratory machinery to marginate towards the vessel wall and cross the endothelial barrier. One regulatory component of neutrophil transendothelial migration is the glycocalyx, which lines the apical endothelial surface. The glycocalyx serves to regulate solute exchange by preventing the movement of large or negatively charged particles (reviewed by [123,124]), however, the depth of the glycocalyx has also been shown to impact PMN adhesion in inflammatory models of pulmonary sepsis [103]. In this model, degradation of the glycocalyx by TNF-α and LPS was shown to promote neutrophil adhesion and enhance tissue damage [103]. Interestingly, circulating MPO has also been shown to reduce the thickness of the endothelial glycocalyx, thus likely indirectly assisting in neutrophil recruitment and transendothelial migration [104]. Given the cationic nature of MPO, it has been shown to bind to the endothelium and promote neutrophil binding by acting as linker between neutrophils and endothelial cells [125]. The ability of MPO to bind biological structures allows for targeted oxidative activity and is likely important in the induction of site-specific effects which may encourage neutrophil recruitment [9,90,125]. In contrast, recent work from our laboratory as well as another independent study demonstrated a negative regulatory role of MPO in neutrophil transendothelial migration [6,126]. Using in vivo intravital imaging and ex vivo adhesion/chemotaxis approaches we found that following neutrophil activation, MPO localization to the neutrophil surface was inhibitory towards neutrophil adhesion to endothelial cells and transendothelial migration [6]. In MPO KO mice or following antibody-mediated inhibition of MPO, the number of neutrophils migrating into inflamed tissue in several models of inflammation was significantly elevated, suggesting a protective or beneficial role for MPO. The above inconsistencies in the underlying MPO function in neutrophil trafficking suggest the potential for context-dependent roles and highlights the need for further investigations into MPO-directed migration.

## 4. MPO Can Serve as a Secondary Messenger to Inform Neighboring Cell Function

MPO can impact the function of many other cells residing in or recruited to inflamed sites. MPO can directly bind platelets [92], fibroblasts [127], endothelial cells [51], and macrophages [105]; however, detailed mechanistic evidence of its impact is still limited and includes both protective and detrimental functions. For example, MPO binding to platelets promotes neutrophil-platelet interactions, stimulating intracellular calcium release and cytoskeletal reorganization, which alters platelet rigidity [91,92]. This has been proposed to promote inflammation and tissue damage. On the other hand, MPO stimulates fibroblast cell migration and promotes the release of collagens I and VI, enhancing tissue regeneration in vivo [127].

Several studies report MPO binding to endothelial cells, and it has even been proposed that endothelial cells may express low levels of MPO or accumulate exogenous MPO [51,56]. Perplexingly, elevated MPO levels are observed in cardiovascular diseases in humans, yet MPO KO animals are more prone to developing atherosclerosis [128]. Further evidence of a beneficial role of MPO in vascular repair has been reported, via MPO-induced activation of ERK, FAK and Akt signaling, and subsequent tube formation [51]. Conversely, MPO inhibition prevents endothelial cell dysfunction in several murine models of vascular inflammation [129,130]. Likewise, neutrophil-derived microparticles containing active MPO exert a cytotoxic effect and damage endothelial cells, suggesting a proinflammatory role for MPO [56]. Finally, similar to endothelial cell studies, we have recently shown that neutrophil-derived microparticles shuttling active MPO destabilize the integrity of epithelial monolayers, resulting in disrupted intestinal epithelial cell proliferation, migration, and overall wound healing processes [5]. Given the evidence presented above, further work is required to dissect the circumstances of seemingly conflicting outcomes of released MPO activity in tissue.

## 5. MPO and Inflammation of the Central Nervous System

Neutrophils have recently emerged as important modulators of neuro-inflammation, specifically impacting the function of vasculature of the brain and immune effector cells of the central nervous system (CNS) (reviewed by [131]). For instance, neutrophil products such as proteinase-3 can degrade endothelial junctions within the cerebral vasculature [132]. Since neutrophils are involved in the maintenance of vascular homeostasis, it is not surprising that MPO has also been implicated in inflammatory conditions of the CNS. For example, neural MPO concentrations are elevated in models of sepsis and several neurodegenerative disorders [133,134]. MPO influences the activity of astrocytes, microglial cells, and neurons by promoting cell death and attenuating neuronal cell responses to stimuli [94]. Additionally, MPO promotes cognitive impairment in murine models of Alzheimer’s disease, as MPO KO animals showed improved performance in behavioral, learning and memory assessments [95].

The catalytic activity of MPO results in the generation of substantial amounts of ROS and other reactive products such as HOCl, and these reactive compounds can impact homeostasis in the vasculature of the brain. For instance, hypothiocyanous acid disrupts the barrier function of brain endothelial cells, and other MPO-derived oxidants cause blood–brain barrier dysfunction [135,136,137]. For these reasons, MPO has been considered as a therapeutic target for several conditions related to inflamed brain tissue (reviewed by [138]). MPO KO mice are resistant to the neuronal cell dysfunction typically observed in murine models of ischemic stroke, and inhibition of MPO with 4-aminobenzoic acid hydrazide (ABAH) protects from tissue damage, suggesting that MPO is a realistic target for therapeutics to mitigate the inflammation associated with brain trauma [108]. Furthermore, MPO inhibition promotes cell proliferation/neurogenesis in these models, and therefore targeting MPO days after the initial injury may provide additional remedial value [139]. Other studies have investigated the efficacy of N-acetyl lysyltyrosylcysteine amide as a therapeutic for treating excess inflammation in the brain [140]. This tripeptide inhibitor of MPO has been shown to limit neuronal damage and inflammation in murine models of stroke and protects bovine aortic endothelial cells from MPO-mediated damage [140,141,142]. It is evident that MPO participates as a regulator of vascular and neuronal cell function in pathologies of the CNS, however further study is warranted to elucidate the specific mechanisms by which MPO directs inflammation in the brain. As described in the sections above, free MPO may possess certain anti-inflammatory benefits, such as inhibition of neutrophil adhesion and transendothelial migration, and so could act to limit tissue injury associated with neutrophil-mediated inflammation, however this has not yet been established in the CNS. The influences of MPO on various biological processes in tissue inflammation are summarized in Figure 1.

## 6. MPO and Cancer

As with tissue inflammation, neutrophils have emerged as important players in the development of many solid tumors. Tumor Associated Neutrophils (TANs) have been shown to be highly plastic, exerting both anti- and pro-tumorigenic activity. Intriguingly, such functional diversity has been suggested to be driven by the cytokine composition of the tumor niche, with TGF-β being one of the major drivers of pro-tumorigenic phenotype acquisition [143]. Alternatively, our group has recently identified temporal differences in the TAN function, where in early/low grade colon tumors TANs exhibited cytotoxic anti-tumorigenic activity, however in advanced CRC, TANs promoted tumor growth. The pro-tumorigenic function of TANs is commonly associated with oxidative burst and damaging ROS generation, to which MPO is a major contributor. Interestingly, MPO interactions with DNA were initially described as protective from oxidative damage [144]. However, it has now been well-established that in fact, MPO-generated ROS and other products such as HOCl potently oxidize and damage DNA, leading to elevated mutational burden and tumor development [145]. HOCl derived from MPO can damage nucleic acids via reactions that form chloramines and nitrogen-centered radicals [146,147,148,149,150]. MPO can also enter the nucleus and directly bind to DNA, as has been shown to occur during the initiation of NETosis to locally promote DNA oxidation [151]. Additionally, MPO can act on various procarcinogens to convert them to their subsequent carcinogenic forms [98]. This is particularly important in cancers related to cigarette smoke and patients with lung cancer, which have elevated MPO levels in lung tissue and serum. MPO-generated HOCl can cause structural changes, strand breaks and impact DNA binding properties, thereby leading to DNA irregularities, which ultimately may promote cancer progression [96]. As mentioned above, other MPO products such as HOBr and HOCl can oxidize components of DNA, resulting in the generation of 5-bromouracil, 5-chlorocytosine and other DNA-centered radicals and this has been proposed for consideration as a potential therapeutic target to prevent MPO-associated carcinogen formation [152,153]. Collectively, these studies demonstrate a major mechanism, where reactive oxidants generated by MPO can promote cancer progression. It is worth noting that a recent work from our laboratory has identified an additional MPO and ROS independent mechanism of PMN-driven DNA damage and genomic instability [154]. We found that extracellular vesicles/microparticles released by activated tissue PMNs shuttled proinflammatory miRNAs to neighboring epithelial cells to downregulate Lamin B1 and RAD-51, respectively destabilizing the nuclear envelope to promote double stranded break (DSB) generation and suppress DSB repair via homologous recombination [154]. Long term activity of these miRNAs potentiated the progression of colorectal cancer [155].

As has been discussed in previous sections, MPO is required for NET release and NETs have been linked to cancer progression in several preclinical models [156]. As with MPO function, NETs have been suggested to both promote and inhibit tumor progression. As such, NET release inhibited metastasis and exhibited cytotoxicity towards melanoma cells in vitro [157]. In contrast, NETs promoted metastasis in models of breast cancer [158], indicating that perhaps the outcome of NET activity is model-dependent and perhaps that of canonical vs. non-canonical MPO function.

Further highlighting the important role of MPO in tumorigenesis, both neutrophils, and specifically MPO have been corelated to tumor size in murine models of lung cancer [159,160]. While neutrophils promoted the development of butylated hydroxytoluene-driven lung carcinogenesis [159], developing tumors in MPO KO mice were significantly smaller compared to tumors in WT animals. In this model, inhibition of MPO with N-acetyl lysyltyrosylcysteine amide also reduced tumor size [160]. Similarly, the addition of exogenous MPO in a murine mammary tumor model stimulated significant increases in tumor size compared to control animals [97]. Authors in that work concluded that the observed impact of MPO on tumor growth was due to its interaction with surrounding fibroblasts to promote collagen release and remodeling of the ECM. In the same report, authors also observed augmented pulmonary metastasis in animals treated with exogenous MPO, providing further evidence for pro-tumorigenic properties of MPO. In contrast, MPO-derived HOCl has been suggested to protect against tumorigenesis by inhibition of NF-κβ and as such, MPO inhibition or in MPO KO mice melanoma growth was augmented [161]. Thus, the impact of MPO on tumor progression is likely multi-faceted, involving numerous signal transduction pathways and more work is warranted to gain a better understanding of the specific role of MPO in cancer.

## 7. Therapeutic Potential of Targeting MPO Activity

MPO targeting as a therapeutic for inflammatory disorders is an emerging area of research (reviewed by [162]). MPO has been advocated as a potential therapeutic target in essentially every inflammatory condition discussed throughout this review. The inhibition of MPO has been proposed as a strategy for resolution of inflammation in the context of IBD [57], atherosclerosis [130,163], neurodegenerative disorders [138,140], cancer [164], and infections [82,165]. While many of these studies show beneficial anti-inflammatory outcomes of MPO inhibition, it is important to remember that MPO can suppress inflammation in certain situations. Additionally, the evidence put forth here regarding the ability of MPO to act as an expansive regulator of inflammation should be carefully considered when attempting pan-inhibition of MPO as a therapeutic strategy. The likelihood for unanticipated off-target consequences is high when one considers the many cell types, processes, and pathways that are impacted by MPO. Furthermore, while designing strategies of MPO inhibition, one cannot disregard the fact that MPO serves a critical role in host defense.

Inhibitors of MPO are broadly classified as irreversible or reversible. Irreversible inhibitors such as the benzoyl hydrazides (eg. ABAH) and 2-thioxanthines act to disrupt the catalytic activity of MPO and have been shown to improve inflammation in several animal models [108,166,167]. For example, the synthetic 2-thioxanthine derivative AZD3241 limits inflammation and reduces the severity of DSS-induced experimental colitis in murine models [76]. Other irreversible 2-thioxanthine inhibitors of MPO, such as AZM198 and AZD5904 show promise as potential therapeutics in a variety of inflammatory models [82,88,101,168]. Reversible inhibitors of MPO include the small peptide N-acetyl lysyltyrosylcysteine amide, which has anti-tumorigenic effects and reduces inflammation in models of stroke and endothelial inflammation [140,142,160]. Many other compounds have been demonstrated to reversibly target MPO, for example hydroxamate compounds [169]. A comprehensive review of clinically relevant MPO inhibitors was recently published [170]. Several MPO inhibitors have gone to clinical trials, however; despite some progress and several assigned patents, there is no MPO inhibitor currently in use in the clinic today. This is likely due to the complex nature of the interactions of MPO with a diverse group of host cells, resulting in distinct physiological effects under different conditions.

Investigations into naturally occurring compounds in plants have revealed several potent inhibitors of MPO, and recently an automated assay to detect MPO inhibition was developed [171]. Alkaloids from the organism *Peganum harmala* can directly bind to the active site of MPO and inhibit its catalytic activity [172]. The plant has long been known to possess anti-inflammatory properties and this study demonstrates that MPO is the target of certain natural remedies [172,173]. Trigonelline, another plant alkaloid, reduces inflammation in rat models of ischemic stroke [174].

Similarly, various microorganisms have developed defensive mechanisms against MPO cytotoxicity. For example, *Staphylococcus aureus* produces a compound known as staphylococcal peroxidase inhibitor (SPIN), which is upregulated following phagocytosis and binds directly to MPO to protect the bacterium from MPO-mediated killing [175,176]. The presence of homologous MPO-inhibiting SPIN proteins was later confirmed in other species of *Staphylococcus*, suggesting that these immune evasion compounds are important factors for bacterial survival [177]. Another example is Propofol, which is a common general anesthetic drug with anti-inflammatory benefits, and which has recently been shown to at least in part exert its anti-inflammatory effect by inhibiting MPO [178,179]. These are just a few examples of natural compounds that impact MPO activity, however, given the exciting therapeutic potential of targeting MPO, many other reversible and irreversible MPO inhibitors have been developed or are currently in development through ongoing work.

In conclusion, the evidence put forth in this review suggest that MPO is an important regulator of host immune system function in the context of tissue inflammation. Ongoing research employing human tissues and MPO-deficient animals has already provided novel insights into how MPO may directly or indirectly impact the function of barrier forming epithelial and endothelial cells as well as various cells of the immune system. MPO inhibition strategies have emerged as interesting and relevant therapeutic approaches towards many inflammatory disorders, however, when designing such therapies, one must consider the multifaceted actions of MPO both within the phagosome (canonical MPO function) and interactions with other cells (the non-canonical MPO activity). Future studies that delve into the unknown mechanisms of MPO function will likely allow for the development of targeted therapeutic strategies to limit excessive inflammation.

## Figures and Tables

**Figure 1 ijms-23-12250-f001:**
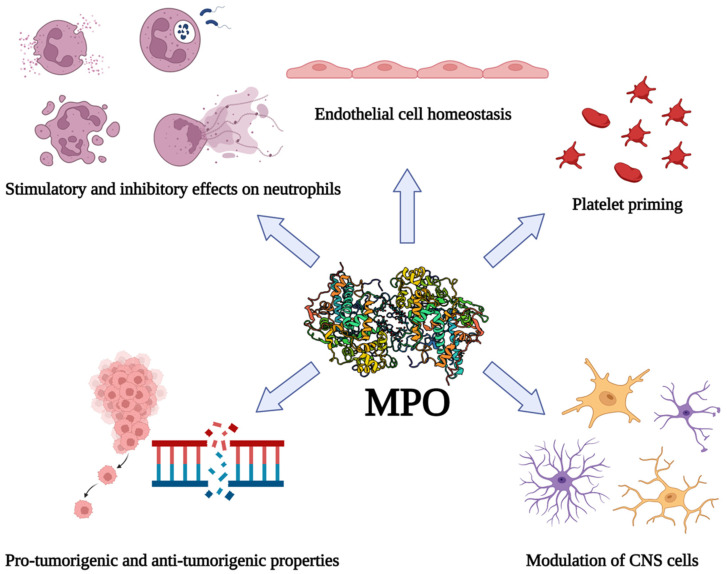
Non-canonical functions of MPO in tissue inflammation. In addition to its canonical antimicrobial activity, MPO can impact many other calls and biological processes in inflammation. Figure generated with Biorender (Publication License #BL24B6KVKU).

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
