# Peer review of "Non-Canonical Functions of Myeloperoxidase in Immune Regulation, Tissue Inflammation and Cancer"

_ijms, 2022, doi:10.3390/ijms232012250_

Round 1
Reviewer 1 Report
This manuscript reports on some of the possible effects induced by myeloperoxidase (MPO) which may be additional to its well established role in the formation of highly reactive oxidants such as HOCl and HOBr, and less reactive species such as HOSCN, nitrogen dioxide radicals and radicals derived from a range of aromatic compounds via the halogenation and peroxidase cycles of the enzyme.
General comments:
The summary is useful, in bringing a wide range of literature reports together in one document, but is rather disappointing in some respects as there is a lack of information or rationale for these additional/alterative (non-canonical) activities. In some cases, these ‘additional’ activities may relate to (unrecognized) oxidant formation effects, but this is not discussed. Inclusion of additional (actual or postulated) mechanisms would be helpful and informative.
Additional specific comments:
11) Line 31: the authors should perhaps use the IUPAC term ‘mass’ instead of ‘weight’
22) Line 38: it should be noted here (and also elsewhere, e.g. line 336 – where MPO damage may be the cause of collagen release and enhanced tumour cell migration) that there is abundant evidence for extracellular damage induced by MPO (see, for example, a recent review on this topic: DOI: 10.1016/j.freeradbiomed.2021.07.007 and the papers referenced within this).
33) Line 53: the oxidase complex assembled in this situation is one of a much larger family, so it is probably advisable to indicate that this is NOX2
44) Line 55: superoxide radical anions
55) Lines 64-66: The topic of MPO binding to pathogens and also many other biological structures is only briefly touched upon in this review, but there is accumulating evidence that this is a critical factor in many of the activities of MPO. This is addressed in some detail in the above review and also many other papers such as papers by Cai (DOI: 10.1038/s41598-019-57299-6) and Klinke (binding to endothelial glycocalyx). There is evidence that such binding can localize oxidation to the site of binding and hence induce site-specific effects.
66) Lines 84-85: references should be included here.
77) Line 109: binding to lipoproteins and ceruloplasmin is also well established, and these may be a major reservoirs and carriers of MPO in the circulation
88) Lines 111-113: binding to RBC is more likely to be via electrostatic interactions with proteins, than via the heme groups present in MPO, as these are buried within the MPO structure.
99) Line 113: albumin is not a highly negatively charged protein – there are many more highly negatively charged species that MPO binds to. The binding to albumin is more likely to be due to the very high abundance of this species.
110) Line 271: N-acetyl (capital ‘N’) – and also at other locations
111) Lines 297-300 and also 305-310: there is a lot of relatively old literature on HOCl (and other MPO-derived species) in inducing DNA and RNA modification, the mechanisms, and the specific sites of alterations: see, for example, work of Prutz (e.g. doi: 10.1006/abbi.1996.0322 ), Whiteman (e.g. https://doi.org/10.1006/bbrc.1999.0448 ), Hawkins (e.g. doi: 10.1021/tx015548d , doi: 10.1021/tx010071r , doi: 10.1042/BJ20020363 ), Masuda (e.g. DOI: 10.1074/jbc.M102700200 )
112) Section 7: mention should perhaps be made of the SPIN proteins discovered by Geisbrecht and colleagues (doi: 10.1073/pnas.1707032114 and later papers)
113) Some of the ‘protective’ actions of MPO with regard to damage may be due to its consumption of H2O2 – see, for example, report by Hawkins doi:10.3390/antiox9121255
Author Response
Reviewer #1 Specific Comments:
We thank the reviewer for the thoughtful comments. In the revised version of the manuscript, we addressed all reviewer’s concerns. All changes are highlighted in red in the revised manuscript for the convenience of the reviewer.
Reviewer Comment 1- Line 31: the authors should perhaps use the IUPAC term ‘mass’ instead of ‘weight’
Author Response 1: This change has been made in the revised manuscript.
Reviewer Comment 2- Line 38: it should be noted here (and also elsewhere, e.g. line 336 – where MPO damage may be the cause of collagen release and enhanced tumour cell migration) that there is abundant evidence for extracellular damage induced by MPO (see, for example, a recent review on this topic: DOI: 10.1016/j.freeradbiomed.2021.07.007 and the papers referenced within this).
Author Response 2: The authors agree with the reviewer that this is important information to include here, and the manuscript has been updated with the suggested reference (and others).
Reviewer Comment 3- Line 53: the oxidase complex assembled in this situation is one of a much larger family, so it is probably advisable to indicate that this is NOX2.
Author Response 3: Per the reviewer’s suggestion, this has been clarified in the text.
Reviewer Comment 4- Line 55: superoxide radical anions.
Author Response 4: Inserted in the revised text.
Reviewer Comment 5 – Lines 64-66: The topic of MPO binding to pathogens and also many other biological structures is only briefly touched upon in this review, but there is accumulating evidence that this is a critical factor in many of the activities of MPO. This is addressed in some detail in the above review and also many other papers such as papers by Cai (DOI: 10.1038/s41598-019-57299-6) and Klinke (binding to endothelial glycocalyx). There is evidence that such binding can localize oxidation to the site of binding and hence induce site-specific effects.
Author Response 5: The authors appreciate this comment. The papers by Cai and Klinke were both included in the original submission. The authors agree that this is an important aspect of MPO activity, so we have included additional information regarding MPO binding biological structures and site-specific oxidative effects in lines 68-71 and 222-226 of the revised manuscript to supplement the citations by Manchanda et al., 2018 and Klinke et al., 2011.
Reviewer Comment 6 – Lines 84-85: references should be included here.
Author Response 6: References have been provided for these statements in the revised manuscript.
Reviewer Comment 7 – Line 109: binding to lipoproteins and ceruloplasmin is also well established, and these may be a major reservoirs and carriers of MPO in the circulation.
Author Response 7: We appreciate this comment by the reviewer. This is an aspect of MPO transport in the circulation that we did not initially consider, and therefore per reviewer’s suggestion have added this along with appropriate references to the revised manuscript.
Reviewer Comment 8 – Lines 111-113: binding to RBC is more likely to be via electrostatic interactions with proteins, than via the heme groups present in MPO, as these are buried within the MPO structure.
Author Response 8: Thank you for the suggestion, the statements in lines 123-128 have been adjusted to address the fact that this is likely due to electrostatic interactions versus direct binding to heme.
Reviewer Comment 9 – Line 113: albumin is not a highly negatively charged protein – there are many more highly negatively charged species that MPO binds to. The binding to albumin is more likely to be due to the very high abundance of this species.
Author Response 9: The speculative statement regarding MPO binding to albumin because of the negative charge on albumin has been removed from the revised manuscript.
Reviewer Comment 10 – Line 271: N-acetyl (capital ‘N’) – and also at other locations
Author Response 10: This has been corrected throughout the manuscript, thank you for catching this error.
Reviewer Comment 11 – Lines 297-300; 305-310: there is a lot of relatively old literature on HOCl (and other MPO-derived species) in inducing DNA and RNA modification, the mechanisms, and the specific sites of alterations: see, for example, work of Prutz (e.g. doi: 10.1006/abbi.1996.0322), Whiteman (e.g. https://doi.org/10.1006/bbrc.1999.0448 ), Hawkins (e.g. doi: 10.1021/tx015548d , doi: 10.1021/tx010071r , doi: 10.1042/BJ20020363 ), Masuda (e.g. DOI: 10.1074/jbc.M102700200)
Author Response 11: Thank you for providing these additional references to include in the revised manuscript. We have expanded the references in these paragraphs (lines 316-318) to include the information as per the suggestion of the reviewer.
Reviewer Comment 12 – Section 7: mention should perhaps be made of the SPIN proteins discovered by Geisbrecht and colleagues (doi: 10.1073/pnas.1707032114 and later papers)
Author Response 12: Thank you for this comment. This is a fascinating mechanism of bacterial immune evasion, and we have updated the manuscript to include a brief description of the SPIN proteins in lines 401-413.
Reviewer Comment 13: Some of the ‘protective’ actions of MPO with regard to damage may be due to its consumption of H2O2 – see, for example, report by Hawkins doi:10.3390/antiox9121255
Author Response 13: The protective effect of MPO via consumption of H2O2 is described in lines 67-71 and lines 160-164 in the context of targeting oxidant production in Salmonella infections. Per the reviewer’s suggestion, we have expanded on this concept to include the modulation of H2O2 cytotoxicity in murine macrophages and included the reference provided in the introduction.
Reviewer 2 Report
The review entitled: “Non-Canonical Functions of Myeloperoxidase in Immune Regulation, Tissue Inflammation and Cancer” by JS Lockhart and R Sumagin submitted to IJMS for publication.
This review deals with the non-canonical functions of MPO and highlights on various disorders but also on the other canonical functions of this key enzyme.
If the authors have stressed on specific actions of MPO in relation within the phagosome, they should also add the important role of nitration reaction and chlorination leading to Cl-Tyr and Nitro-Tyr which can impact signaling pathways.
Another important aspect that is still lacking amongst the either the non-canonical function of MPO in the therapeutic of targeting MPO activity. For doing, the authors should be taken into account the role of stem cells as already reported elsewhere (see references below). Figure 1 can be revised by taking into account this aspect.
Franck T et al (2021). Muscle Derived Mesenchymal Stem Cells Inhibit the Activity of the Free and the Neutrophil Extracellular Trap (NET)-Bond Myeloperoxidase. Cells. 2021 Dec 10;10(12):3486. doi: 10.3390/cells10123486. PMID: 34943996; PMCID: PMC8700239.
Odobasic D and Holdsworth SR (2021) Emerging Cellular Therapies for Anti-myeloperoxidase Vasculitis and Other Autoimmune Diseases. Front. Immunol. 12:642127. doi: 10.3389/fimmu.2021.642127
Overall, the review presented here is very interesting but needs to be completed with additional data (e.g. stem cells) for the benefit of the readers of IJMS. Therefore, a major revision is required. Once everything will be fine, there is no reason to accept this interesting review for publication in IJMS.
Author Response
Reviewer #2 Specific Comments:
We thank the reviewer for the thoughtful comments. In the revised version of the manuscript we addressed all reviewer’s concerns. All changes are highlighted in red in the revised manuscript for convenience of the reviewer.
Reviewer Comment 1: If the authors have stressed on specific actions of MPO in relation within the phagosome, they should also add the important role of nitration reaction and chlorination leading to Cl-Tyr and Nitro-Tyr which can impact signaling pathways.
Author Response 1: We agree that this is an important aspect of MPO activity. Per reviewer suggestion, we discussed and referenced this (Prutz, 1996; Masuda et al., 2001; Whiteman et al., 1999; Hawkins and Davies, 2001; Hawkins and Davies, 2002) in lines 324-329 of the revised manuscript. We appreciate that these references describe the formation of nitrogen-centered radicals and chloramines to impact nucleic acids, and therefore we have included additional information in the introduction to address the role of HOCl in the chlorination of proteins/ peptides (lines 76-80).
Reviewer Comment 2: Another important aspect that is still lacking amongst the either the non-canonical function of MPO in the therapeutic of targeting MPO activity. For doing, the authors should be taken into account the role of stem cells as already reported elsewhere (see references below). Figure 1 can be revised by taking into account this aspect.
Franck T et al (2021). Muscle Derived Mesenchymal Stem Cells Inhibit the Activity of the Free and the Neutrophil Extracellular Trap (NET)-Bond Myeloperoxidase. Cells. 2021 Dec 10;10(12):3486. doi: 10.3390/cells10123486. PMID: 34943996; PMCID: PMC8700239.
Odobasic D and Holdsworth SR (2021) Emerging Cellular Therapies for Anti-myeloperoxidase Vasculitis and Other Autoimmune Diseases. Front. Immunol. 12:642127. doi: 10.3389/fimmu.2021.642127
Overall, the review presented here is very interesting but needs to be completed with additional data (e.g. stem cells) for the benefit of the readers of IJMS. Therefore, a major revision is required. Once everything will be fine, there is no reason to accept this interesting review for publication in IJMS.
Author Response 2: We thank the reviewer for this comment. The interactions of stem cells with MPO is an intriguing and new avenue of research. Per the reviewer’s suggestion, we included discussion on stem cell interactions/inhibition of MPO, along with the suggested references in lines 91-100 of the revised manuscript. As for including MSCs as a part of Figure 1, we felt that this is a bit premature. The figure aims to illustrate the direct impact of MPO on established biological processes, and its impact on MSC physiology although novel and of great interest is only emerging.
Round 2
Reviewer 2 Report
Dear Editor,
The authors have correctly answered to my concerns. Even though they still reserved to include Stem cells in the general figure 1 to establish the direct interaction with the MPO enzyme. Nevertheless, I recommend that this interesting work can be published.